# Combating Adversaries with Anti-Adversaries

**Motasem Alfarra** [1]  **Juan C. Pérez** [1]  **Ali Thabet** [1]  **Adel Bibi** [2]  **Philip H.S. Torr** [2]  **Bernard Ghanem** [1]

## Abstract

Deep neural networks are vulnerable to small input perturbations known as adversarial attacks. Inspired by the fact that these adversaries are constructed by iteratively minimizing the confidence of a network for the true class label, we propose the anti-adversary layer, aimed at countering this effect. In particular, our layer generates an input perturbation in the opposite direction of the adversarial one, and feeds the classifier a perturbed version of the input. Our approach is training-free and theoretically supported. We verify the effectiveness of our approach by combining our layer with both nominally and robustly trained models, and conduct large scale experiments from black-box to adaptive attacks on CIFAR10, CIFAR100 and ImageNet. Our anti-adversary layer significantly enhances model robustness while coming at no cost on clean accuracy.

## 1. Introduction

Deep Neural Networks (DNNs) are vulnerable to small input perturbations known as adversarial attacks (Szegedy et al., 2013; Goodfellow et al., 2015). While there has been a proliferation in the literature aimed at training DNNs that are robust to adversarial attacks, assessing the robustness of such defenses remains an elusive task. This difficulty is due to the following reasons. (**i**) The robustness of models varies according to the information an attacker is assumed to know, *e.g.* training data, gradients, logits, *etc.*, which, for ease, dichotomously categorizes adversaries as being black- or white-box. Consequently, this categorization results in difficulties when comparing defenses tailored to a specific type of adversaries. For instance, several defenses crafted for robustness against white-box adversaries were later broken with their weaker black-box counterparts (Papernot et al., 2016; Brendel et al., 2018). (**ii**) In addition,

[1]King Abdullah University of Science and Technology (KAUST) [2]University of Oxford. Correspondence to: Motasem Alfarra <motasem.alfarra@kaust.edu.sa>.

*Accepted by the ICML 2021 workshop on A Blessing in Disguise: The Prospects and Perils of Adversarial Machine Learning.* Copyright 2021 by the author(s).

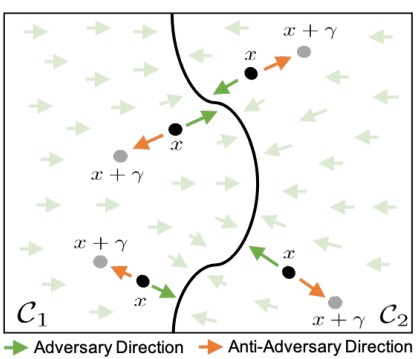

Figure 1. **Anti-adversary classifier.** The flow field of adversarial perturbations is shown in light green for both classes $\mathcal{C}_1$ and $\mathcal{C}_2$. The anti-adversary we construct pulls a given point $x$ to $(x + \gamma)$ by moving in the direction *opposite* to that of the adversary flow.

robustness, as evaluated empirically, can be overestimated if fewer efforts are invested in *adaptively* constructing a stronger attack (Tramer et al., 2020; Carlini et al., 2019). The lack of reliable assessments has been responsible for a false sense of security, as several thought-to-be-strong defenses against white-box adversaries were later broken with better carefully-crafted adaptive attacks (Athalye et al., 2018a). The few defenses that have stood the test of time usually come at the expense of costly training and performance degradation on clean samples (Tsipras et al., 2019). Even worse, while most of these defenses are meant to resist white-box attacks, little effort has been invested into resisting the black-box counterparts, which are the more common and practical ones (Byun et al., 2021), *e.g.* online APIs such as IBM Watson and Microsoft Azure tend not to disclose information about the inner workings of their models.

In this work, we propose a simple, generic, training-free layer that improves the robustness of both nominally and robustly trained models. Specifically, given a base classifier $f : \mathbb{R}^n \to \mathcal{Y}$, which maps $\mathbb{R}^n$ to labels in the set $\mathcal{Y}$, and an input $x$, our layer constructs a data- and model-dependent perturbation $\gamma$ in the *anti-adversary* direction, *i.e.* the direction that maximizes the base classifier's confidence on the pseudo-label $f(x)$, as illustrated in Figure 1. The new sample $(x + \gamma)$ is then fed to the base classifier $f$ in lieu of $x$. We dub this complete approach as the *anti-adversary* classifier $g$. By conducting an extensive robustness assessment of our classifier $g$ on several datasets and under the

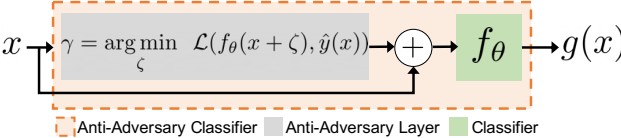

$$x \rightarrow \boxed{\gamma = \arg\min_{\zeta} \mathcal{L}(f_\theta(x+\zeta), \hat{y}(x)) \rightarrow \bigoplus \rightarrow \boxed{f_\theta}} \rightarrow g(x)$$

☐ Anti-Adversary Classifier  ▨ Anti-Adversary Layer  ▨ Classifier

*Figure 2.* **The Anti-Adversary classifier.** Our anti-adversary layer generates $\gamma$ for each $x$ and $f_\theta$, and feeds $(x + \gamma)$ to $f_\theta$ resulting into our anti-adversary classifier $g$).

## 1.1. Related Work

Given the security concerns that adversarial vulnerability brings, a stream of works was developed to build models that are not only accurate but also robust against adversarial attacks. From the black-box perspective, several defenses have shown their effectiveness to defend against such attacks (Rakin et al., 2018; Liu et al., 2017; Dong et al., 2020). Moreover, SND (Byun et al., 2021) showed that small input perturbations can enhance the robustness of pretrained models against black-box attacks. However, the main drawback of randomized methods is that they can be bypassed with Expectation Over Transformation (EOT) (Athalye et al., 2018b). Once an attacker accesses the gradients, *i.e.* grey-box attackers, the robust accuracy of such defenses drastically decreases. Adversarial training (Madry et al., 2018) and its enhanced version (Hendrycks et al., 2019; Carmon et al., 2019) are among the most effective defenses against such attacks. This was further improved by incorporating additional regularizers such as TRADES (Zhang et al., 2019) and MART (Wang et al., 2019), or combining adversarial training with network pruning as in HYDRA (Sehwag et al., 2020), or perturbing network parameters (Wu et al., 2020).

## 2. Methodology

**Motivation.** Adversary directions are the ones that maximize a loss function in the input, *i.e.* move an input $x$ closer to the decision boundary, resulting in minimizing the prediction's confidence on the correct label. In this work, we leverage this fact by prepending a layer to a trained model to generate a new input $(x + \gamma)$, which moves $x$ far from the decision boundary, thus hindering the capacity of attackers to successfully tailor adversaries.

## 2.1. Preliminaries and Notation

We use $f_\theta : \mathbb{R}^n \rightarrow \mathcal{P}(\mathcal{Y})$ to denote a classifier, *e.g.* a neural network, parameterized by $\theta$, where $\mathcal{P}(\mathcal{Y})$ refers to a probability simplex over the set $\mathcal{Y} = \{1, 2, \ldots, k\}$ of $k$ labels. For an input $x$, an attacker constructs a small perturbation $\delta$

---

**Algorithm 1** Anti-adversary classifier $g$

**Function** `AntiAdversaryForward`$(f_\theta, x, \alpha, K)$:
  **Initialize:** $\gamma^0 = 0$
  $\hat{y}(x) = \arg\max_i f_\theta^i(x)$
  **for** $k = 0 \ldots K - 1$ **do**
    $\gamma^{k+1} = \gamma^k - \alpha \, \text{sign}(\nabla_{\gamma^k} \mathcal{L}(f_\theta(x + \gamma^k), \hat{y}))$
  **end**
  **return** $f_\theta(x + \gamma^K)$

---

(*e.g.* $\|\delta\|_p \leq \epsilon$) such that $\arg\max_i f_\theta^i(x + \delta) \neq y$, where $y$ is the true label for $x$. In particular, one popular approach to constructing $\delta$ is by solving the following constrained problem with a suitable loss function $\mathcal{L}$:

$$\max_\delta \ \mathcal{L}(f_\theta(x + \delta), y) \qquad \text{s.t. } \|\delta\|_p \leq \epsilon. \qquad (1)$$

Depending on the information about $f_\theta$ made available to the attacker when solving Problem (1), the adversary $\delta$ can generally be categorized into one of three types. (**i**) **Black-box:** Only function evaluations $f_\theta$ are available when solving (1). (**ii**) **Grey-box:** Only $f_\theta$ and $\nabla_x f_\theta$ are accessible when solving (1), with no other intermediate layer representations or intermediate gradients available to the attacker. (**iii**) **Adaptive:** The attacker has full knowledge about the classifier $f_\theta$ for solving (1), including the parameters $\theta$, intermediate-layer gradients, training data, *etc.*

## 2.2. Anti-Adversary Layer

Analogous to the procedure used for constructing an adversary by solving (1), we propose, given a classifier, to prepend a layer that perturbs the input so as to maximize the classifier's prediction confidence at this input, hence the term *anti-adversary*. Formally, given a classifier $f_\theta$, our proposed anti-adversary classifier $g$ (prepending $f_\theta$ with an anti-adversary layer) is given as follows:

$$\begin{aligned} g(x) &= f_\theta(x + \gamma), \\ \text{s.t. } \gamma &= \arg\min_\zeta \ \mathcal{L}(f_\theta(x + \zeta), \hat{y}(x)), \end{aligned} \qquad (2)$$

where $\hat{y}(x) = \arg\max_i f_\theta^i(x)$ is the predicted label. Note that our proposed anti-adversary classifier $g$ is agnostic to the choice of $f_\theta$. Moreover, it does not require retraining $f_\theta$, unlike previous works (Xie et al., 2018; Byun et al., 2021) that add random perturbations to the input, further hurting clean accuracy. This is because instances that are correctly classified by $f_\theta$, *i.e.* instances where $y = \arg\max_i f_\theta^i(x)$, will be (by construction as per optimization (2)) classified correctly by $g$. As such, our anti-adversary layer only increases the confidence of the top prediction of $f_\theta(x)$. We illustrate our approach in Figure 2. Finally, our anti-adversary layer solves Problem (2) with $K$ signed gradient descent iterations, zero initialization, and $\mathcal{L}$ being the cross-entropy loss. Algorithm 1 summarizes the forward pass of $g$.

*Table 1.* **Robustness of nominally trained models against black-box attacks:** We present the robustness of a nominally trained model against Bandits and NES, and how this robustness enhances when equipping the model with SND (Byun et al., 2021) and our anti-adversary layer (Anti-Adv). We perform all attacks with both $5k$ and $10k$ queries. Results shown are accuracy measured in % where bold numbers correspond to best performance. Our approach outperforms SND by a significant margin across datasets, attacks, and number of queries.

| | CIFAR10 | | | | | ImageNet | | | | |
| | Clean | Bandits | | NES | | Clean | Bandits | | NES | |
| | | 5K | 10K | 5K | 10K | | 5K | 10K | 5K | 10K |
|---|---|---|---|---|---|---|---|---|---|---|
| Nominal Training | 93.7 | 24.0 | 17.2 | 5.8 | 4.8 | 79.2 | 65.2 | 58.2 | 22.4 | 21.0 |
| + SND (Byun et al., 2021) | 92.9 | 84.5 | 84.3 | 30.3 | 25.5 | 79.2 | 72.8 | 73.2 | 65.4 | 60.2 |
| + Anti-Adv | 93.7 | **85.5** | **86.4** | **77.0** | **72.7** | 79.2 | **73.6** | **74.4** | **67.2** | **66.0** |

## 3. Experiments

We validate the effectiveness of our proposed anti-adversary classifier $g$ by evaluating robustness under a wide spectrum of adversaries. (**i**) We compare the robustness of $f_\theta$ against our anti-adversary classifier $g$ against popular black-box attacks (Bandits (Ilyas et al., 2019), NES (Ilyas et al., 2018) and Square (Andriushchenko et al., 2020)) both when $f_\theta$ is nominally and robustly trained. We observe significant robustness improvements over $f_\theta$ with virtually no drop in clean accuracy, while also outperforming recently proposed defenses, such as SND (Byun et al., 2021). (**ii**) We experiment in the more challenging grey-box setting with AutoAttack (Croce & Hein, 2020a) (in particular with the strong attacks APGD, ADLR (Croce & Hein, 2020a), and FAB (Croce & Hein, 2020b)), when $f_\theta$ is trained robustly with TRADES (Zhang et al., 2019), ImageNet-Pre (Hendrycks et al., 2019), MART (Wang et al., 2019), HYDRA (Sehwag et al., 2020), and AWP (Wu et al., 2020). In all experiments, we do *not* retrain $f_\theta$ after prepending our anti-adversary layer. We set $K = 2$ and $\alpha = 0.15$ whenever Algorithm 1 is used, unless stated otherwise.

### 3.1. Robustness under Black-Box Attacks

We study the robustness gains against black-box attacks of prepending our anti-adversary layer to a classifier $f_\theta$. This setting can be of interest for commercially-available APIs, *e.g.* BigML, that only allow access to model predictions and so can only be targeted with black-box adversaries.

**Robustness when $f_\theta$ is Nominally Trained.** We conduct experiments with ResNet18 (He et al., 2016) on CIFAR10 (Krizhevsky et al., 2009) and ResNet50 on ImageNet (Krizhevsky et al., 2012). We compare the clean and robust accuracies of $f_\theta$ against SND (Byun et al., 2021), a recently proposed approach for robustness through input randomization, and our anti-adversary classifier $g$. We attack all methods with two black-box attacks, Bandits and NES, with query budgets of $5k$ and $10k$, respectively. For this experiment, we set $\alpha = 0.01$ in Algorithm 1. Following common practice (Byun et al., 2021), we compute all accuracies on 1000 and 500 instances of CIFAR10 and ImageNet, respec-

tively, and report results in Table 1.

As shown in Table 1, nominally trained models $f_\theta$ are not robust: while their clean accuracies on CIFAR10 and ImageNet are $93.7\%$ and $79.2\%$, respectively, these drop to $4.8\%$ and $21\%$ under black-box attacks. Moreover, while SND improves $f_\theta$'s robustness significantly, *i.e.* to $25.5\%$ and $60.2\%$ on CIFAR10 and ImageNet, our proposed anti-adversary outperforms SND across attacks, query budgets, and datasets. For instance, under the limited $5k$ query budget, our anti-adversary classifier outperforms SND by $1\%$ and $46.7\%$ on CIFAR10 against Bandits and NES attacks. The robustness improvements over SND broadens when the attacks are granted a larger $10k$ budget. For instance, on ImageNet under such budget, our anti-adversary outperforms SND by $1.2\%$ and $5.8\%$ for Bandits and NES attacks, respectively, while coming at no cost to clean accuracy.

**Robustness when $f_\theta$ is Robustly Trained.** The previous section provided evidence that our anti-adversary layer can improve black-box robustness of a nominally trained $f_\theta$. Here, we investigate whether our anti-adversary layer can also improve robustness in a more challenging setting when $f_\theta$ is robustly trained. This setup is of interest if $f_\theta$ was trained robustly against white-box attacks and then deployed as a service, and thus only function evaluations are available to the attacker (1), and so only black-box robustness is of essence. We thus study black-box robustness with our proposed anti-adversary layer over five state-of-the-art robustly trained $f_\theta$ (TRADES, IN-Pret, MART, HYDRA, and AWP) on both the CIFAR10 and CIFAR100 datasets. Similar to the previous experimental setup, we report robust accuracy for 1000 instances of the test set under the Bandits and NES black-box attacks. However, for the more efficient Square attack, we report robust accuracy on the full test set.

In Tables 2, we report the robust test accuracies on CIFAR10 and CIFAR100, respectively, under black-box attacks, highlighting the strongest attack in grey. Confirming our previous observations, prepending our anti-adversary layer to $f_\theta$ has no impact on clean accuracy. More importantly, although $f_\theta$ is robustly trained, and thus has high robust accuracy, our proposed anti-adversary layer can still boost

Table 2. **Black-box attacks on robust models equipped with Anti-Adv.** We report clean (%) and robust accuracies against *Bandits*, *NES* and *Square attack* on CIFAR10 and CIFAR100. Bold values indicate highest accuracy in each experiment. Our layer provides across-the-board improvements on robustness against all attacks, without affecting clean accuracy.

| CIFAR10 | Clean | Bandits | NES | Square |
|---|---|---|---|---|
| TRADES | 85.4 | 64.7 | 74.7 | 53.1 |
| + Anti-Adv | 85.4 | **84.6** | **83.0** | **71.7** |
| ImageNet-Pre | 88.7 | 68.4 | 78.1 | 62.4 |
| + Anti-Adv | 88.7 | **88.1** | **86.4** | **78.5** |
| MART | 87.6 | 72.0 | 79.5 | 64.9 |
| + Anti-Adv | 87.6 | **86.5** | **85.3** | **78.0** |
| HYDRA | 90.1 | 69.8 | 79.2 | 65.0 |
| + Anti-Adv | 90.1 | **89.4** | **87.7** | **78.8** |
| AWP | 88.5 | 71.5 | 80.1 | 66.2 |
| + Anti-Adv | 88.5 | **87.4** | **86.9** | **80.7** |

| CIFAR100 | Clean | Bandits | NES | Square |
|---|---|---|---|---|
| ImageNet-Pre | 59.0 | 40.6 | 47.7 | 34.6 |
| + Anti-Adv | 58.9 | **58.2** | **55.3** | **42.4** |
| AWP | 59.4 | 39.8 | 47.3 | 34.7 |
| + Anti-Adv | 59.4 | **57.7** | **53.8** | **46.4** |

robustness by an impressive ∼ 15%. For instance, even for the highest worst-case robust accuracy on CIFAR10 (66.18% achieved by AWP), the anti-adversary improves robustness by 14.53% to reach 80.71%. Similarly, for CIFAR100 our proposed anti-adversary layer improves the worst-case black-box robustness of AWP by 11.7%. Overall, our anti-adversary layer consistently improves black-box robust accuracy against all attacks and for all robust training methods $f_\theta$ on both CIFAR10 and CIFAR100.

**SND + Robustly Trained $f_\theta$.** While SND (Byun et al., 2021) does not report performance when used on top of robust models, we conduct experiments with AWP equipped with SND. We observe that SND significantly affects AWP both in terms of clean and robust accuracies. For instance, when AWP is equipped with SND its clean accuracy drops from 88.5% to 70.03%, and its accuracy against Square attacks drops from 66.18% to 59.12%. These results suggest that our proposed anti-adversary layer is superior to SND.

### 3.2. Robustness under Grey-Box Attacks

In this setting, the attacker (1) can only access function evaluations of the classifier and its gradients w.r.t. the input. That is, the attacker ignores the classifier's inner workings and its training specifications. While this setup is less realistic than the black-box setting, it is an interesting measure of robustness by providing more information to the attacker,

Table 3. **Grey-box attacks on robust models equipped with Anti-Adv.** We report clean and robust accuracies against *APGD*, *ADLR*, and *AutoAttack* (AA) on CIFAR10 and CIFAR100.

| CIFAR10 | Clean | APGD | ADLR | AA |
|---|---|---|---|---|
| TRADES | 84.92 | 55.31 | 53.12 | 53.11 |
| + Anti-Adv | 84.88 | **77.20** | **77.05** | **71.71** |
| ImageNet-Pre | 87.11 | 57.65 | 55.32 | 55.31 |
| + Anti-Adv | 87.11 | **78.76** | **79.02** | **76.01** |
| MART | 87.50 | 62.18 | 56.80 | 56.75 |
| + Anti-Adv | 87.50 | **81.07** | **80.54** | **76.76** |
| HYDRA | 88.98 | 60.13 | 57.66 | 57.64 |
| + Anti-Adv | 88.95 | **80.37** | **81.42** | **76.39** |
| AWP | 88.25 | 63.81 | 60.53 | 60.53 |
| + Anti-Adv | 88.25 | **80.65** | **81.47** | **79.21** |

| CIFAR100 | Clean | APGD | ADLR | AA |
|---|---|---|---|---|
| ImageNet-Pre | 59.37 | 33.45 | 29.03 | 28.96 |
| + Anti-Adv | 58.42 | **47.63** | **45.29** | **40.68** |
| AWP | 60.38 | 33.56 | 29.16 | 29.15 |
| + Anti-Adv | 60.38 | **44.21** | **40.32** | **39.57** |

which is related to why most prior works report performance in this category by computing accuracy under PGD (Madry et al., 2018) or AutoAttack, such as (Xie et al., 2019).

Similar to the previous section, we prepend our proposed anti-adversary layer to robustly trained classifiers $f_\theta$ and assess robustness on both CIFAR10 and CIFAR100. We report robust accuracy against grey-box gradient-based attacks, namely APGD, ADLR and FAB, and measure the accuracy under AutoAttack (the worst-case accuracy across these attacks) with $\epsilon = {}^8/_{255}$ in (1). We underscore that AutoAttack is currently the strongest attack in this setting and so is used as a standard to benchmark defenses.

In Table 3 we report the robust accuracy on CIFAR10 and CIFAR100, respectively, and highlight the strongest attack in grey. We first observe that our anti-adversary layer improves robust accuracy by an impressive ∼ 19% on average against AutoAttack. In particular, for the strongest defense we consider, AWP, adversarial robustness increases from 60.53% to an astounding 79.21%. We observe similar results for CIFAR100. Table 3 shows that the anti-adversary layer adds an average improvement of ∼ 11%, where the adversarial robustness of ImageNet-Pre increases from 28.96% to over 40%. The improvement is consistent across all defenses on CIFAR100 with a worst-case drop in clean accuracy of 1%. In addition, integrating SND with AWP comes at a notable drop in clean accuracy (from 88.25% to 70.03%) along with a drastic drop in robust accuracy (from 60.53% to 27.04%) under AutoAttack on CIFAR10.

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

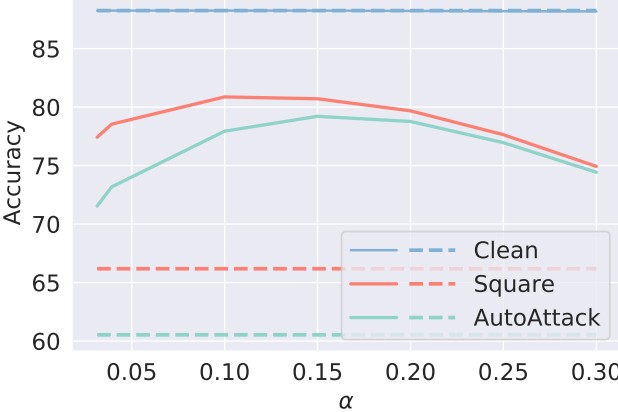

*Figure 3.* **Effect of varying $\alpha$ on clean and robust accuracy for AWP+Anti-Adv on CIFAR10.** Dashed lines represent the performance of AWP. Our layer provides notable improvements on robust accuracy against both Square attack and AutoAttack with different choices of $\alpha$ and with no effect on clean accuracy.

## A. Extra Experiments

### A.1. Robustness under Adaptive Attacks: Worst-Case Performance

In this setting, we analyze the worst-case robustness of our proposed anti-adversary classifier $g$. In particular, and under the least realistic setting, we assume that our anti-adversary classifier $g$ is fully transparent to the attacker (1) when tailoring an adversary. Following the recommendations of (Tramer et al., 2020), we explore several directions to construct an attack such as Expectation Over Transformation (EOT) (Athalye et al., 2018b; Tramer et al., 2020). Since our anti-adversary layer is deterministic as we always use zero initialization in Algorithm 1, improving gradient estimate with EOT is ineffective. However, we observe that the anti-adversary layer depends on the pseudo-label of the perturbation-free instance $x$ produced by $f_\theta$, *i.e.* $\hat{y}(x) = \arg\max_i f_\theta^i(x)$. Therefore, an attacker with access to the internal structure of $g$ can first design an adversary $\delta$ such that $\hat{y}(x + \delta) \neq y$ with $\|\delta\|_p \leq \epsilon$ following (1) where $y$ is the true label for $x$. If the perturbation $\delta$ is constructed in this way, it will cause both $f_\theta$ and $g$ to produce different predictions for $x$ and $(x + \delta)$. This implies that, in the least realistic adversary setting, the set of adversaries that fools $f_\theta$ fools $g$ as well. Accordingly, we argue that the worst-case robust accuracy for $g$ under adaptive attacks is lower bounded by the robust accuracy of the underlying classifier $f_\theta$. While, as noted in previous sections, our anti-adversary layer boosts robust accuracy over all tested datasets and classifiers $f_\theta$ (nominally or robustly trained), the worst-case robustness under the least realistic setting (adaptive attacks) is lower bounded by the robustness of $f_\theta$. This highlights our motivation that prepending our layer is of a great value to existing robust models due to its simplicity and having no cost on clean accuracy.

### A.2. Ablations

Our proposed Algorithm 1 has two main parameters, namely the learning rate $\alpha$ and the number of iterations $K$. We ablate both to see their impact on the robustness gains. All experiments are conducted, when $f_\theta$ is robustly trained with AWP. First, we fix $K = 2$ and vary $\alpha$ from the set $\{8/255, 10/255, 0.1, 0.15, 0.2, 0.25, 0.3\}$. In Figure 3, we compare $f_\theta$ to our anti-adversary classifier $g$ in terms of clean and robust accuracies under a black-box (Square) and a grey-box (AutoAttack) attacks. We observe that the effect of varying $\alpha$ on clean accuracy is negligible (almost non-existent), as shown in blue. On the other hand, while the robust accuracy varies with $\alpha$, the robustness gain of $g$ over $f_\theta$ is always $\geq 10\%$ over all considered $\alpha$. Next, we consider the same setup but study the effect of varying $K \in \{1, 2, 3\}$ while fixing $\alpha = 0.15$. Results in Figure 4 show that all choices of $K$ lead to significant enhancement in model robustness against all attacks, with $K = 3$ performing best. This confirms our claim at the end of Section 2 that the better the solver for (2), the better the robustness performance of our anti-adversary classifier. Note that while one could further improve the robustness gains by increasing $K$, this improvement comes at the expense of more computations. It is worthwhile to mention that the cost of computing the anti-adversary is $(K + 1)$ forward and $K$ backward passes which is marginal for small values of $K$.

**Experiments with $K = 1$.** For completeness, we conduct both black-box (using Square attack) and grey-box with the

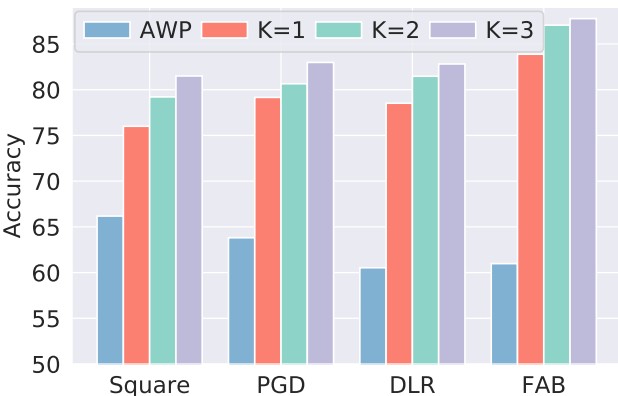

*Figure 4.* **Effect of varying** $K$ **on robust accuracy for AWP+Anti-Adv on CIFAR10.** The better the solver for (2) is, the bigger the robustness gains that our layer provides.

*Table 4.* **Equipping robustly trained models with Anti-Adv on CIFAR10 against black- and grey-box attacks.** We report clean accuracy (%) and robust accuracy against *APGD*, *ADLR*, *FAB*, *Square* and *AutoAttack* where bold numbers correspond to highest accuracy in each experiment. The last column summarizes the improvement on the AutoAttack benchmark.

| | Clean | APGD | ADLR | FAB | Square | AutoAttack | Improvement |
|---|---|---|---|---|---|---|---|
| ImageNet-Pre | 87.11 | 57.65 | 55.32 | 55.69 | 62.39 | 55.31 | - |
| + Anti-Adv ($\alpha = {}^{10}/_{255}$) | 87.11 | 61.92 | 59.06 | 73.93 | 69.01 | 58.77 | 3.46 |
| + Anti-Adv ($\alpha = 0.15$) | 87.02 | **77.74** | **75.09** | **81.25** | **75.72** | **72.63** | **17.32** |
| MART | 87.50 | 62.18 | 56.80 | 57.34 | 64.87 | 56.75 | - |
| + Anti-Adv ($\alpha = {}^{10}/_{255}$) | 87.50 | 70.98 | 65.03 | 77.15 | 75.47 | 64.51 | 7.73 |
| + Anti-Adv ($\alpha = 0.15$) | 87.29 | **75.67** | **72.90** | **79.69** | **70.00** | **67.42** | **10.67** |
| HYDRA | 88.98 | 60.13 | 57.66 | 58.42 | 65.01 | 57.64 | - |
| + Anti-Adv ($\alpha = {}^{10}/_{255}$) | 88.98 | 71.84 | 69.35 | 83.72 | 76.87 | 68.98 | 11.34 |
| + Anti-Adv ($\alpha = 0.15$) | 88.93 | **78.55** | **78.27** | **84.36** | **75.98** | **73.59** | **15.95** |
| AWP | 88.25 | 63.81 | 60.53 | 60.98 | 66.18 | 60.53 | - |
| + Anti-Adv ($\alpha = {}^{10}/_{255}$) | 88.25 | 70.86 | 68.80 | 82.06 | 75.39 | 68.57 | 8.04 |
| + Anti-Adv ($\alpha = 0.15$) | 88.10 | **79.16** | **78.52** | **83.88** | **76.00** | **74.47** | **13.34** |

cheapest version of our Algorithm 1 where we set $K = 1$. In this experiment, we vary the learning rate $\alpha \in \{{}^{10}/_{255}, 0.15\}$ reporting the results on both CIFAR10 and CIFAR100. As shown in Tables 4 and 5, our anti-adversary layer even with $K = 1$ provides a remarkable improvement on network robustness against both black-box and grey-box attacks. In particular, we improve the robust accuracy against the strongest black-box attack (shaded in grey), Square attack, by at least 5% and 6% on CIFAR10 and CIFAR100, respectively. This improvement extends to cover the grey-box settings as well, where the improvement against AutoAttack is 13% on CIFAR10 and 5% on CIFAR100.

**Experiments with** $K \in \{4, 5\}$**.** For completeness, we analyze the effect of enlarging the number of iterations $K$ used to solve Equation (2). In Figure 5, we show the robust accuracy of AWP when combined with our anti-adversary layer when varying $K \in \{2, 3, 4, 5\}$ with $\alpha = 0.15$. We observe that the larger the number of iterations used, the larger the robustness gains that our layer brings.

## B. Implementation Details.

**Nominally trained models.** For CIFAR10 models, we trained ResNet18 from scratch for 90 epochs with SGD with an initial learning rate of 0.1, momentum of 0.9, and weight decay of $2 \times 10^{-4}$. We multiply the step size by 0.1 after every 30 epochs. For ImageNet experiments, we used pretrained weights of ResNet50 from PyTorch (Paszke et al., 2019). For all

*Table 5.* **Equipping robustly trained models with Anti-Adv on CIFAR100 against black- and grey-box attacks.** Similarly to CI-FAR10 experiments, our layer provides a sizable improvement to robustness without sacrificing clean accuracy.

|  | Clean | APGD | ADLR | FAB | Square | AutoAttack | Improvement |
|---|---|---|---|---|---|---|---|
| ImageNet-Pre | 59.37 | 33.45 | 29.03 | 29.34 | 34.55 | 28.96 | - |
| + Anti-Adv ($\alpha = {}^{10}/_{255}$) | 59.29 | 34.91 | 30.87 | 39.46 | 39.44 | 30.61 | 1.65 |
| + Anti-Adv ($\alpha = 0.15$) | 59.24 | **35.55** | **31.56** | **40.86** | **40.76** | **31.34** | **2.38** |
| AWP | 60.38 | 33.56 | 29.16 | 29.48 | 34.66 | 29.15 | - |
| + Anti-Adv ($\alpha = {}^{10}/_{255}$) | 60.38 | 34.30 | 30.17 | 38.09 | 36.88 | 30.15 | 1.00 |
| + Anti-Adv ($\alpha = 0.15$) | 60.38 | **39.16** | **35.30** | **47.18** | **44.30** | **34.88** | **5.73** |

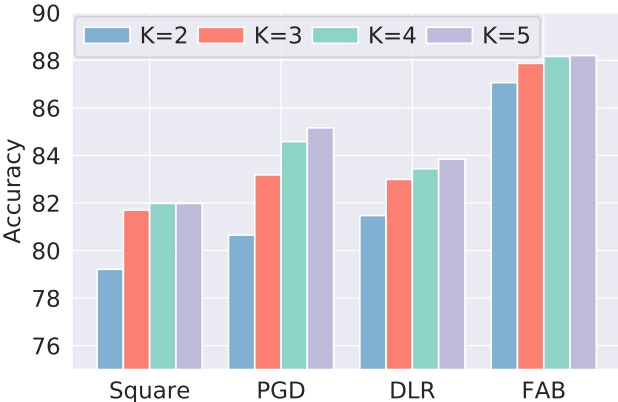

*Figure 5.* **Effect of varying $K$ on robust accuracy for AWP+Anti-Adv on CIFAR10.** The better the solver for (2) is, the bigger the robustness gains that our layer provides.

robust models, we used provided weights by the respective authors.

**Black-box attacks.** We used NES and Bandits from https://raw.githubusercontent.com/MadryLab/blackbox-bandits/master/src/main.py , while we used the Square attack from the AutoAttack repo at https://github.com/fra31/auto-attack.

**Grey-box attacks.** We used APGD, ADLR, FAB and the worst case accuracy AutoAttack from the aforementioned AutoAttack repository.