# OpenReview forum: "Combating Adversaries with Anti-Adversaries"
_ICML.cc/2021/Workshop/AML — ICML 2021 Workshop AML Poster_

### Official Review · Reviewer_fFTS · 2021-06-20
**More experiments of white-box attack defense is required.**

**Rating:** Accept
**Confidence:** 3

**Review:**

This paper introduces an anti-adversary layer in the network structure, which is equivalent to changing the network structure and pulling data points away from the classification decision boundary, The paper proves that it can effectively resist black box, gray box and adaptive attacks.
Experiments are carried out on various network structures and datasets, and the effectiveness of the proposed method is proved.
I have a question:
The defensive approach to changing the network structure seems to have universal defense effect regardless of the attack method. The article does not discuss the defensive nature of the proposed defense method against the white box attack. White box attack is a kind of attack in which network structure can be obtained. Is the general defense strength obtained by changing network structure also effective against white box attack? The paper seems to lack sufficient experiment and analysis.

---

### Decision · Program_Chairs · 2021-06-21

**Decision:**

Accept (Poster)

**Comment:**

This paper proposed an anti-adversary layer in the network structure. Experiments are carried out on various network structures and datasets, and the effectiveness of the proposed method is proved. Some concerns are raised by the reviewer, which can be further addressed.